# Influence of the Timing of Leptomeningeal Metastasis on the Outcome of *EGFR*-Mutant Lung Adenocarcinoma Patients and Predictors of Detectable *EGFR* Mutations in Cerebrospinal Fluid

**DOI:** 10.3390/cancers14122824

**Published:** 2022-06-07

**Authors:** Pei-Ya Liao, Wei-Fan Ou, Kang-Yi Su, Ming-Hsi Sun, Chih-Mei Huang, Kun-Chieh Chen, Kuo-Hsuan Hsu, Sung-Liang Yu, Yen-Hsiang Huang, Jeng-Sen Tseng, Tsung-Ying Yang, Gee-Chen Chang

**Affiliations:** 1Division of Chest Medicine, Department of Internal Medicine, Taichung Veterans General Hospital, Taichung 407, Taiwan; pat790328@vghtc.gov.tw (P.-Y.L.); spmanner@vghtc.gov.tw (W.-F.O.); a13628@mail.cmuh.org.tw (C.-M.H.); tyyang@vghtc.gov.tw (T.-Y.Y.); 2Department of Clinical Laboratory Sciences and Medical Biotechnology, College of Medicine, National Taiwan University, Taipei 100, Taiwan; suky@ntu.edu.tw (K.-Y.S.); slyu@ntu.edu.tw (S.-L.Y.); 3Department of Laboratory Medicine, National Taiwan University Hospital, Taipei 100, Taiwan; 4Department of Neurological Institute, Taichung Veterans General Hospital, Taichung 407, Taiwan; mhsun@vghtc.gov.tw; 5Division of Pulmonary Medicine, Department of Internal Medicine, Chung Shan Medical University Hospital, Taichung 402, Taiwan; ckjohn@mail2000.com.tw (K.-C.C.); cshy1888@csmu.edu.tw (G.-C.C.); 6School of Medicine, Chung Shan Medical University, Taichung 402, Taiwan; 7Institute of Medicine, Chung Shan Medical University, Taichung 402, Taiwan; 8Division of Critical Care and Respiratory Therapy, Department of Internal Medicine, Taichung Veterans General Hospital, Taichung 407, Taiwan; khhsu@vghtc.gov.tw; 9Institute of Medical Device and Imaging, College of Medicine, National Taiwan University, Taipei 100, Taiwan; 10Graduate Institute of Pathology, College of Medicine, National Taiwan University, Taipei 100, Taiwan; 11Graduate Institute of Clinical Medicine, College of Medicine, National Taiwan University, Taipei 100, Taiwan; 12Institute of Biomedical Sciences, National Chung Hsing University, Taichung 402, Taiwan; 13College of Medicine, National Yang Ming Chiao Tung University, Taipei 112, Taiwan; 14Department of Post-Baccalaureate Medicine, College of Medicine, National Chung Hsing University, Taichung 402, Taiwan; 15Department of Life Sciences, National Chung Hsing University, Taichung 402, Taiwan

**Keywords:** epidermal growth factor receptor (EGFR), lung adenocarcinoma, leptomeningeal metastasis, cerebrospinal fluid (CSF)

## Abstract

**Simple Summary:**

Leptomeningeal metastasis (LM) is a devastating complication of lung cancer, with a generally poor outcome. We conduct the present study to evaluate the association between clinical presentations, brain images, tumor cell counts of the cerebrospinal fluid (CSF), and the *epidermal growth factor receptor* (*EGFR*) mutation detection rate in CSF among *EGFR*-mutant lung adenocarcinoma patients with LM and accessed the influence of the timing of LM occurrence on patient outcomes. Tumor cell numbers were semi-quantified according to tumor cells per high power field of CSF cytological slides. Radiological burden was assessed using a four-point scoring system, which evaluated LM-involved areas on brain magnetic resonance imaging. Our results suggest the association between the radiological severity score of LM, CSF tumor cell counts, and *EGFR* mutation detection rate in CSF. Furthermore, LM prior to first-line EGFR-tyrosine kinase inhibitor treatment was associated with an independently worse outcome.

**Abstract:**

Background: We aim to evaluate the influence of the timing of leptomeningeal metastasis (LM) occurrence on the outcome of *EGFR*-mutant lung adenocarcinoma and to explore the predictors of detectable *EGFR* mutation in the cerebrospinal fluid (CSF). Methods: *EGFR*-mutant lung adenocarcinoma patients with cytologically confirmed LM were included for analysis. *EGFR* mutation in CSF was detected by MALDI-TOF MS plus PNA. Results: A total of 43 patients was analyzed. Of them, 8 (18.6%) were diagnosed with LM prior to first-line EGFR-TKI treatment (early onset), while 35 patients (81.4%) developed LM after first-line EGFR-TKI treatment (late onset). Multivariate analysis suggested that both late-onset LM (aHR 0.31 (95% CI 0.10–0.94), *p* = 0.038) and a history of third-generation EGFR-TKI treatment (aHR 0.24 (95% CI 0.09–0.67), *p* = 0.006) independently predicted a favorable outcome. *EGFR* mutation detection sensitivity in CSF was 81.4%. The radiological burden of LM significantly correlated with CSF tumor cell counts (*p* = 0.013) with higher CSF tumor cell counts predicting a higher detection sensitivity of *EGFR* mutation (*p* = 0.042). Conclusions: Early onset LM was an independently poor prognostic factor. A higher radiological severity score of LM could predict higher tumor cell counts in CSF, which in turn were associated with a higher detection rate of *EGFR* mutation.

## 1. Introduction

Lung cancer is the leading cause of cancer-related death worldwide [1]. During the recent decade, the treatment of advanced stage non-small cell lung cancer (NSCLC) has been directed more towards personalized therapy. Both histological classification and driver mutation status play a critical role in the decisions that are made regarding treatment [2]. *Epidermal growth factor receptor* (*EGFR*) is one of the most common driver mutations of lung cancer. Among Asians diagnosed with lung adenocarcinoma, approximately 50–60% of patients harbor an *EGFR* mutation [3,4] and could benefit from EGFR-tyrosine kinase inhibitor (TKI) therapy [5,6].

Central nervous system (CNS) metastases are a common complication seen in lung cancer patients. The incidence of CNS metastases in lung cancer patients is 10–20% at diagnosis and could be as high as 40–50% during the disease course [7]. CNS metastases generally imply a worse response to systemic treatment and a shorter survival time [8]. CNS metastases involve metastases to the brain parenchyma, dura, and leptomeninges. Of the three, leptomeningeal metastasis (LM) is associated with both a much worse outcome and more neurological symptoms [9,10]. Importantly, lung cancer patients harboring an *EGFR* mutation possess a higher risk of CNS metastases [11,12].

Owing to their smaller molecular weight and higher penetration rate through the blood–brain barrier, intracranial metastatic lesions are more likely to respond to TKI than to conventional chemotherapy. Additionally, EGFR-TKI could lengthen the overall survival of *EGFR*-mutant NSCLC patients with LM [13,14]. When LM occurs prior to EGFR-TKI treatment, patients reserve the opportunity to control LM through targeted therapy. By contrast, when LM occurs after EGFR-TKI treatment, patients may experience a better quality of life and a longer disease control period from the initial EGFR-TKI treatment. It remains unknown whether the timing of LM occurrence influences patient outcome.

Cerebrospinal fluid (CSF) can be used as a liquid biopsy specimen for the assessment of genetic alterations. However, the *EGFR* mutation detection rate seems lower in CSF specimens [15,16]. Prior studies suggested that positive neurological symptoms and evidence of LM on brain magnetic resonance imaging (MRI) could predict a higher detection rate of *EGFR* mutations in CSF, and the severity of LM as seen on MRI was associated with patient outcome [16,17]. However, whether the severity of LM on brain MRI correlates with the number of tumor cells in CSF and the *EGFR* mutation detection rate remains uncertain. We conduct the present study in order to evaluate the association between clinical presentations, brain images, tumor cell counting of CSF, and *EGFR* mutation detection rates among *EGFR*-mutant lung adenocarcinoma patients with LM. Moreover, we accessed the influence of LM timing on patient outcomes.

## 2. Materials and Methods

### 2.1. Patients

This is a retrospective cohort study, in which we analyzed lung cancer patients diagnosed and treated at Taichung Veterans General Hospital from January 2013 to August 2019. To be eligible for the study, patients were required to have cytologically or pathologically confirmed lung adenocarcinoma, cytologically confirmed LM, detectable *EGFR* mutation, had EGFR-TKI as the first-line treatment, available CSF specimens for semi-quantification of tumor cell counts and *EGFR* mutation testing, had a brain MRI for LM severity assessment, and precise clinical follow-up data. Patients were excluded if they had mixed components of other histological types, other active malignancies, or incomplete data records.

### 2.2. Data Records and Response Evaluation

Clinical data for analysis included patient age, gender, smoking status, the Eastern Cooperative Oncology Group Performance Status (ECOG PS), driver mutation status, tumor stage, treatment regimens and response, timing of LM occurrence, clinical presentations of LM, and survival status. Lung cancer TNM (tumor, node, and metastases) staging was conducted according to the 8th edition of the American Joint Committee on Cancer (AJCC) staging system [18]. In the case of the timing of LM occurrence, we defined LM before and after first-line EGFR-TKI treatment as early onset and late onset, respectively. This study was approved by the Institutional Review Board of Taichung Veterans General Hospital (IRB Nos. CF12019, CF20175 and CF20176). Written informed consent for clinical data records and genetic testing was obtained from all patients.

### 2.3. Severity of Leptomeningeal Metastasis on Brain MRI Scans

The severity of LM was assessed by a four-point scoring system, which was modified from the study performed by Nevel KS et al. and evaluates whether LM has involved the four areas of the brain, including cerebrum, ventricle, brainstem, and cerebellum, as seen on MRI scans [17]. All the brain images of our patients were reviewed by one neurosurgeon and one neuroradiologist.

### 2.4. Semi-Quantification of CSF Tumor Cells

Samples for CSF cytology were centrifuged using the Thermo Scientific Shandon Cytospin^®^ 4 (Shandon Scientific Ltd., Cheshire, UK). Two layers of cell smear samples were prepared and then fixed in 95% alcohol for 30 min. The slides were stained using Liu’s stain following the manufacturer’s instructions. We designed a semi-quantification scale to evaluate the tumor cell numbers in the CSF specimens. All slides were reviewed by a senior cytologist, with tumor cell numbers counted according to tumor cells per high power field (400×). We defined the tumor cells as negative, trace, 1+, 2+, and 3+ if the average tumor cell counts were 0, less than one, one, two, and three or more tumor cells per high-power field, respectively.

### 2.5. EGFR Mutation Assay

*EGFR* mutations of extracranial specimens were assessed using matrix-assisted laser desorption ionization-time of flight mass spectrometry (MALDI-TOF MS) [3]. Briefly, we performed the testing according to the instructions provided by the MassARRAY system (Sequenom, San Diego, CA, USA). With respect to the biochemical reaction, polymerase chain reaction was used to amplify the region containing the tyrosine kinase domain of the *EGFR* exons 18, 19, 20, and 21. A single nucleotide extension was then performed by primers and corresponding detection probes to amplify the region containing each target mutation. After SpectroClean Resin clean up, the samples were loaded onto the matrix of SpectroCHIP by Nanodispenser (Matrix) and then analyzed by Bruker Autoflex MALDI-TOF MS. Data were collected and analyzed by Typer4 software (Sequenom). All the tests were performed by ISO15189-certified TR6 Pharmacogenomics Lab, National Research Program for Biopharmaceuticals (NRPB), at the National Center of Excellence for Clinical Trial and Research of National Taiwan University Hospital. The *EGFR* mutation status of the CSF specimens was determined by a combination of MALDI-TOF MS and peptide nucleic acid (PNA), which can enrich the mutant alleles and enhance detection sensitivity as we previously described [19].

### 2.6. Statistical Methods

The Fisher’s exact test was used to evaluate the association between categorized variables, while the Kaplan–Meier method was used to estimate the survival time. Overall survival was determined as the time from initiation of first-line EGFR-TKI treatment to death by any cause. The association between LM severity as seen on brain MRI and intracranial pressure was accessed by the Student’s *t*-test. Differences in survival time were analyzed by the log-rank test. A Cox proportional hazard model was performed for multivariate analyses of the risk factors of survival outcomes. All statistical tests were carried out using SPSS 15.0 (SPSS Inc., Chicago, IL, USA). Two-tailed tests and *p*-values < 0.05 for significance were implemented.

## 3. Results

### 3.1. Patients and Their Demographic Data

A total of 43 patients was included for analysis. The patient characteristics are summarized in Table 1. The median age was 59 years. Of them, 20 patients (46.5%) were female, and 30 patients (69.8%) had never smoked. The baseline ECOG PS was 0–2 in 28 patients (65.1%). Exon 19 deletions (30.2%) and exon 21 L858R (51.2%) accounted for the most common *EGFR* mutation types. The first-line treatments involved gefitinib, erlotinib, and afatinib in 11 (25.6%), 16 (37.2), and 16 (37.2%) patients, respectively. In the case of the timing of LM occurrence, 8 patients (18.6%) had LM prior to first-line EGFR-TKI treatment (early onset), while 35 patients (81.4%) developed LM after first-line EGFR-TKI treatment (late onset).

### 3.2. Detection of EGFR Mutation in CSF Specimens

The timing and results of *EGFR* mutation detection in CSF specimens are summarized in Figure 1. Among patients with early onset LM, a CSF specimen was obtained prior to EGFR-TKI treatment in one patient, while specimens of the other seven patients were obtained after EGFR-TKI treatment. A total of 35 patients (81.4%) revealed detectable *EGFR* mutations in their CSF specimens. With regard to the sensitive mutations, all 35 patients had compatible mutation types to that of their baseline *EGFR* mutations. With regard to T790M mutation, a total of 42 specimens was obtained after progression to first-line EGFR-TKI, with only 8 (19.0%) showing positive T790M mutation; 3 had compatible results with other re-biopsy sites. Among the other 34 patients, 5 had positive T790M mutation from other re-biopsy sites.

### 3.3. Clinical Presentation, Extents of LM on Brain MRI, and the Yield Rate of EGFR Mutation in CSF

The results of the association of severity between LM on brain MRI with clinical presentation and the yield rate of *EGFR* mutation in CSF are summarized in Figure 2. The scores of LM severity on brain MRI were 0, 1, 2, 3, and 4 in 4, 10, 18, 10, and 1 patient(s), respectively. With regard to the clinical presentations, there was no significant association of LM severity with seizure, neurological deficits, or intracranial pressure (*p* = 1.000, 0.160, and 0.283, respectively).

In the case of the semi-quantification of CSF tumor cells, the cell counts were trace, 1+, 2+, and 3+ in 13, 20, 9, and 1 patient(s), respectively. A higher LM severity score as seen on brain MRI (2–4 vs. 0–1) was associated with a higher possibility of having ≥1+ tumor cells in CSF (82.9% vs. 42.0%, *p* = 0.013). Moreover, CSF tumor cells ≥ 1+ were associated with a higher CSF *EGFR* mutation detection rate (90.0% vs. 61.5%, *p* = 0.042). In the case of T790M mutation, a numerically higher positive rate was also observed in patients with CSF tumor cells ≥ 1+ (24.1% vs. 7.7%), although the *p*-value was not significant.

### 3.4. Onset of Leptomeningeal Metastasis and Its Impact on Outcomes

The comparisons of patient characteristics and treatments between patients with early onset and late-onset LM are shown in Table 2. There were no significant differences in demographic data, *EGFR* mutation types, severity of LM, the presence of hydrocephalus, bevacizumab therapy, or subsequent third-generation EGFR-TKI therapy. Of note, patients with early onset LM were more likely to undergo erlotinib treatment as their first-line EGFR-TKI therapy, with no patients being treated with gefitinib (*p* = 0.006). Moreover, these patients were less likely to receive whole-brain radiotherapy for LM (*p* = 0.017).

After the exclusion of six patients who did not have any measurable lesions, the objective response rate of first-line EGFR-TKI treatment was 73.0%. The median progression-free survival and overall survival periods were 10.9 months (95% CI 7.2–14.6) and 28.0 months (95% CI 16.9–39.2), respectively. The results of univariate analysis regarding overall survival with patient characteristics, treatments, and the timing of LM occurrence are shown in Table 3 and Figure 3. Of note, history of third-generation EGFR-TKI treatment (HR 0.19 (95% CI 0.08–0.46), *p* < 0.001), emergence of T790M mutation (HR 0.30 (95% CI 0.11–0.82), *p* = 0.018), and late onset of LM (HR 0.31 (95% CI 0.13–0.73), *p* = 0.008) were all associated with longer overall survival. The overall survival period of patients with late onset and early onset LM was 29.9 months (95% CI 20.4–39.4) and 12.3 months (95% CI 4.2–20.4), respectively (log-rank, *p* = 0.005).

In the multivariate analysis, a history of third-generation EGFR-TKI treatment (aHR 0.24 (95% CI 0.09–0.67), *p* = 0.006) and late-onset LM (aHR 0.31 (95% CI 0.10–0.94), *p* = 0.038) both independently predicted a favorable outcome. In the stratified analysis, among the patients without any history of third-generation EGFR-TKI treatment, late-onset LM was still associated with a longer overall survival (*p* < 0.001) (Figure 3B). Among the patients with a history of third-generation EGFR-TKI treatment, late-onset LM was associated with a numerically longer overall survival period (40.5 months (95% CI 35.8–45.2) vs. 15.0 months (95% CI NR-NR)). Of them, there were only two patients with early onset LM that had a history of third-generation EGFR-TKI treatment, with a not significant *p*-value. Moreover, a numerically shorter progression-free survival of first-line EGFR-TKI treatment was observed in patients with early onset LM (6.8 months (95% CI 0.0–15.1) vs. 11.1 months (95% CI 5.4–15.7), *p* = 0.261).

## 4. Discussion

LM is a devastating complication resulting from lung cancer, with a generally poor outcome, and a median survival time limited to only a few months. Owing to the current advances in lung cancer management and improvements in survival, the incidence of LM has increased [20]. However, LM could be the initial presentation of lung cancer in a small set of patients [21]. Although many studies have investigated the prognostic factors of these patients, such as their clinical characteristics, driver mutation status, CSF features, and treatment strategies, the influence of the timing of LM occurrence remains uncertain. Although *EGFR*-mutant lung adenocarcinoma patients with early onset LM reserve the opportunity to control their LM through targeted therapy, our results suggest that early onset LM independently predicts a poor outcome.

As most lung cancer clinical trials only enroll patients with stable and asymptomatic CNS metastasis, and people with LM often have multiple neurological symptoms, these patients have usually been excluded from clinical trials. Therefore, options for exploring what the optimal therapy is for LM are limited. Currently, there is no clear consensus on the optimal treatment for lung cancer patients with LM. The role of radiotherapy, antiangiogenic therapy, and intrathecal chemotherapy remains uncertain [22]. Similar conditions have been observed in our population, as the regimen of first-line EGFR-TKI, bevacizumab therapy, and whole-brain radiotherapy did not influence overall survival. In LM with *EGFR* mutations, pulse erlotinib and afatinib, osimertinib, and other novel EGFR-TKIs with a higher BBB penetration rate have shown clinical benefits [23,24]. Moreover, a retrospective study that enrolled 304 *EGFR*-mutant NSCLC patients with a propensity-matched analysis suggested that osimertinib could reduce the incidence of LM [25]. In addition to the timing of LM occurrence, a history of third-generation EGFR-TKI therapy is also an independently favorable prognostic factor. Although the emergence of T790M was a good prognostic factor in univariate analysis, its influence was not significant in the multivariate analytic model. These findings are compatible with those of prior studies, which have suggested that the benefits of osimertinib are evident, regardless of T790M status [26,27]. When we determined the survival time from the first LM detection, the survivals of patients with early onset and late-onset LM were 12.5 months (95% CI 4.6–20.4) and 8.5 months (95% CI 4.3–12.6), respectively. Although the difference is not statistically significant (*p* = 0.489), patients with late-onset LM experienced a numerically shorter survival from LM occurrence to death, which implies limited treatment options for patients suffering LM after the initial EGFR-TKI treatment.

Among lung cancer patients experiencing LM, CSF cell-free DNA could be a useful liquid biopsy specimen for identifying genetic alterations and guiding subsequent therapy [28,29]. Both single-gene analysis through polymerase-chain-reaction-based methods or multiplex detection by next-generation sequencing are practicable methods. However, the major challenge clinicians will face is the limited number of tumor cells and genomic materials taken from CSF, which in turn would lead to a lower detection sensitivity [15,16]. Moreover, studies that link the association of the severity of LM in brain images with CSF tumor cell counts and *EGFR* detection rates are limited. In a retrospective study, Nevel et al. applied a quantitative assessment of LM disease burden on brain MRI and suggested that the extent of radiographic involvement, as well as quantification of CSF tumor cells and cell-free DNA, were all associated with patient outcomes [17]. The radiological burden of LM is scored by the number of LM involvements in eight locations: the cerebrum, cerebellum, ventricle, brainstem, cranial nerves, cervical spinal cord, thoracic spinal cord, and lumbosacral spinal cord. Patients with involvements at ≥3 MRI sites experienced a significantly shorter survival time. A similar condition was observed in patients with CSF tumor cells ≥50/mL and cell-free DNA >0.02 ng/mL. However, the driver mutation status of these patient was heterogeneous and the association of radiological burden with CSF tumor cells and cell-free DNA was not clearly described.

In the present study, we modified the scoring system regarding the extent of LM on MRI because the anatomic locations of the cranial nerves and brainstem are very close, and MRI of the spinal cord is not usually available for every patient. In this study, we did not observe any significant correlation between radiological LM severity and clinical presentation. However, a higher radiological severity score for LM could predict higher tumor cell counts in CSF, which would be associated with a higher detection rate of *EGFR* mutation. These findings are compatible with liquid biopsy, which uses plasma cell-free DNA, of which, a higher tumor burden could predict a higher detection sensitivity [30].

Only 19.0% of our patients (n = 8) showed detectable T790M in CSF after progression to EGFR-TKI treatment, which was much lower than prior studies [31,32]. Among the other 34 patients, 5 had positive T790M mutation from other re-biopsy sites. Similar to the conditions observed in liquid biopsy, which used plasma specimens, it remains more challenging to detect a T790M mutation, which leads to a lower detection sensitivity when compared with other activating *EGFR* mutations [33,34]. Another possibility is that some patients experience CNS progression due to inadequate CNS penetration of EGFR-TKI, rather than the emergence of the altered drug target [35].

The major limitations of this study are its retrospective nature and the limited patient numbers. Although the data were collected retrospectively, we tried to ensure the validity of patients’ characteristics, treatment course and genetic alterations, as well as quantification of radiological and cytological findings. In addition to the association between the timing of LM occurrence and patient outcome, we attempted to explore the correlation between the radiological severity of LM, CSF tumor cell counts, and *EGFR* detection sensitivity, which could provide meaningful information in clinical practice.

## 5. Conclusions

Early onset LM predicted a significantly worse outcome in *EGFR*-mutant lung adenocarcinoma patients and radiological severity predicted higher tumor cell counts if CSF and a higher detection rate of *EGFR* mutation in CSF.

## Figures and Tables

**Figure 1 cancers-14-02824-f001:**
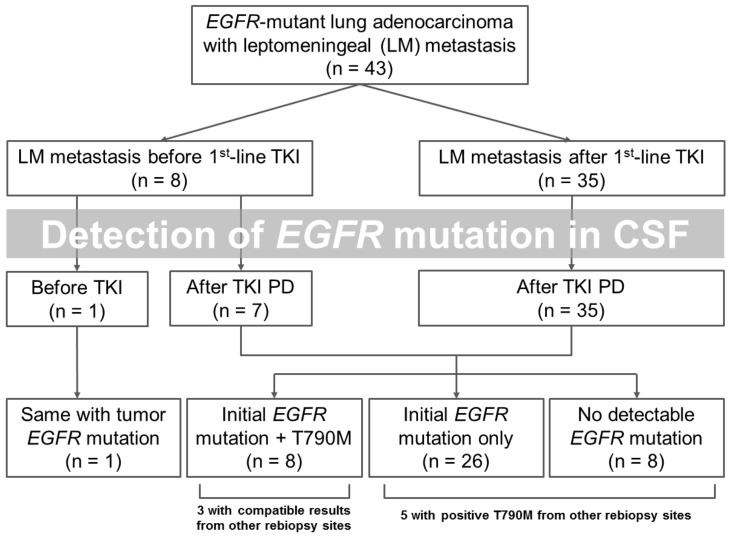
Timing and results of *EGFR* mutation detection in CSF specimens.

**Figure 2 cancers-14-02824-f002:**
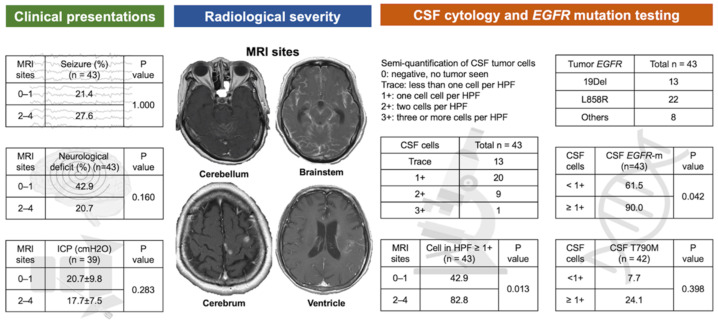
Clinical presentation, extent of leptomeningeal metastasis on brain MRI, and the yield rate of *EGFR* mutation in cerebrospinal fluids (CSF).

**Figure 3 cancers-14-02824-f003:**
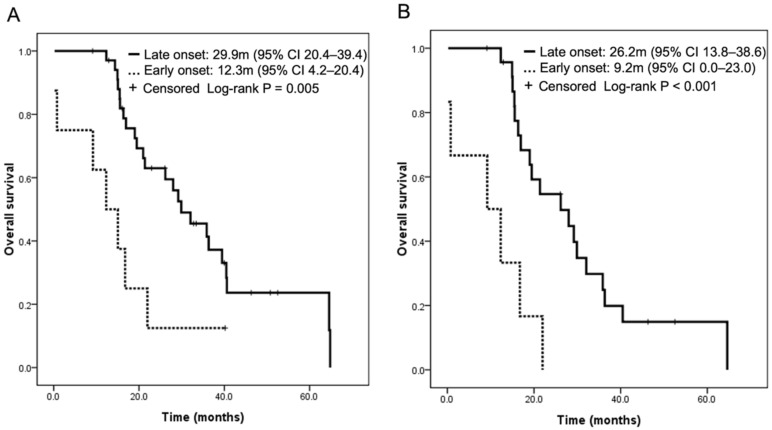
The onset of leptomeningeal metastasis and overall survival of EGFR-TKI treatment: overall population (**A**) and those without third-generation EGFR-TKI treatment (**B**).

**Table 1 cancers-14-02824-t001:** Demographic data and patient characteristics.

Characteristics	*n* = 43
Age, years, median (range)	59 (41–83)
Gender, n (%)	
Female	20 (46.5)
Male	23 (53.5)
Smoking status, n (%)	
Never smoked	30 (69.8)
Smokers	13 (30.2)
ECOG PS, n (%)	
0–2	28 (65.1)
3–4	15 (34.9)
Baseline *EGFR* mutations, n (%)	
Exon 19 deletions (19Del)	13 (30.2)
Exon 21 L858R	22 (51.2)
Others *	8 (18.6)
First-line treatment, n (%)	
Gefitinib	11 (25.6)
Erlotinib	16 (37.2)
Afatinib	16 (37.2)
Onset of leptomeningeal metastasis, n (%)	
Before first-line EGFR-TKI (early onset)	8 (18.6)
After first-line EGFR-TKI (late onset)	35 (81.4)

ECOG PS, Eastern Cooperative Oncology Group Performance Status; EGFR, epidermal growth factor receptor; TKI, tyrosine kinase inhibitor. * Includes 2 with L861Q, 2 with G719X + 19Del, 3 with G719X + S768I, and 1 with L858R + S768I.

**Table 2 cancers-14-02824-t002:** Univariate analysis of patient characteristics, treatment patterns, and the onset of leptomeningeal metastasis.

Characteristics	Early Onset, n (%)	Late Onset, n (%)	*p* Value *
Age			0.656
≥65 years	1 (12.5)	10 (28.6)	
<65 years	7 (87.5)	25 (71.4)	
Gender			0.440
Female	5 (62.5)	15 (42.9)	
Male	3 (37.5)	20 (57.1)	
Smoking status			1.000
Never smoked	6 (75.0)	24 (68.6)	
Smokers	2 (25.0)	11 (31.4)	
ECOG PS			1.000
0–2	5 (62.5)	23 (65.7)	
3–4	3 (37.5)	12 (34.3)	
*EGFR* mutation			0.885
19Del	2 (25.0)	11 (31.4)	
L858R	5 (62.5)	17 (48.6)	
Others	1 (12.5)	7 (20.0)	
First-line EGFR-TKI			0.006
Gefitinib	0 (0.0)	11 (31.4)	
Erlotinib	7 (87.5)	9 (25.7)	
Afatinib	1 (12.5)	15 (42.9)	
Third-G TKI therapy ^#^			0.132
Yes	2 (25.0)	20 (57.1)	
No	6 (75.0)	15 (42.9)	
Bevacizumab			1.000
Yes	2 (25.0)	11 (31.4)	
No	6 (75.0)	24 (68.6)	
WBRT			0.017
Yes	1 (12.5)	22 (62.9)	
No	7 (87.5)	13 (37.1)	

ECOG PS, Eastern Cooperative Oncology Group Performance Status; EGFR, epidermal growth factor receptor; TKI, tyrosine kinase inhibitor; WBRT, whole-brain radiotherapy; MRI, magnetic resonance imaging. * By Fisher’s exact test. ^#^ Includes one with CO-1686; otherwise, all with osimertinib treatment.

**Table 3 cancers-14-02824-t003:** Univariate analysis of the overall survival among *EGFR*-mutant patients with leptomeningeal metastasis.

Characteristics	Hazard Ratio	95% CI	*p*-Value *
Age ≥ 65 vs. < 65 years	1.25	0.53–2.94	0.616
Female vs. male	1.40	0.68–2.90	0.366
Smokers vs. never smoked	0.54	0.22–1.28	0.161
ECOG PS: 3–4 vs. 0–2	1.26	0.60–2.65	0.552
*EGFR* 19Del vs. others	0.71	0.32–1.57	0.402
First-G TKI vs. Second-G TKI	0.75	0.35–1.60	0.451
Third-G TKI use: yes vs. no ^#^	0.19	0.08–0.46	<0.001
Bevacizumab use: yes vs. no	0.84	0.38–1.84	0.660
WBRT: yes vs. no	0.58	0.28–1.21	0.149
Emergence of T790M: yes vs. no ^&^	0.30	0.11–0.82	0.018
Extent of metastasis: 2–4 vs. 0–1	0.88	0.61–1.28	0.506
Hydrocephalus: yes vs. no	1.18	0.56–2.49	0662
Late onset vs. early onset	0.31	0.13–0.73	0.008

ECOG PS, Eastern Cooperative Oncology Group Performance Status; EGFR, epidermal growth factor receptor; TKI, tyrosine kinase inhibitor; WBRT, whole-brain radiotherapy. * By Cox regression analysis model. ^#^ Includes one with CO-1686; otherwise, all with osimertinib treatment. ^&^ Either from CSF or other re-biopsy site(s).

## Data Availability

The datasets used and/or analyzed during the current study are available from the corresponding author upon reasonable request.

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
