# Peer review of "Influence of the Timing of Leptomeningeal Metastasis on the Outcome of EGFR-Mutant Lung Adenocarcinoma Patients and Predictors of Detectable EGFR Mutations in Cerebrospinal Fluid"

_cancers, 2022, doi:10.3390/cancers14122824_

Round 1

Reviewer 1 Report

The manuscript by Liao Pei-Ya et al titled as “Influence of the timing of leptomeningeal metastasis on outcome of EGFR-mutant lung adenocarcinoma patients and predictors of detectable EGFR mutations in cerebrospinal fluid” investigates the association between radiological severity of LM, CSF tumor cell numbers and EGFR mutation detection rate among the lung-adenocarcinoma patients. The findings could be important in the therapeutic strategy. The manuscript is well written, and the data have been presented appropriately. However, this study has been performed on a small number of patients, which is a major limiting factor to make the conclusions. The authors should discuss the associated limitations. Also, authors should add methods and the data on EGFR mutation detection (MALDI-TOF MS). It can be provided as a supplementary data. 

Author Response

Reviewer 1:

  1. The manuscript by Liao Pei-Ya et al. titled as “Influence of the timing of leptomeningeal metastasis on outcome of EGFR-mutant lung adenocarcinoma patients and predictors of detectable EGFR mutations in cerebrospinal fluid” investigates the association between radiological severity of LM, CSF tumor cell numbers and EGFR mutation detection rate among the lung-adenocarcinoma patients. The findings could be important in the therapeutic strategy. The manuscript is well written, and the data have been presented appropriately.

Response: We appreciate the reviewer’s encouragement.

  1. However, this study has been performed on a small number of patients, which is a major limiting factor to make the conclusions. The authors should discuss the associated limitations.

Response: We agree with the reviewer’s comments. We’ve mentioned that the major limitation of current study is its retrospective nature and limited patient numbers, and associated descriptions are made in page 10, line 330-336.

  1. Also, authors should add methods and the data on EGFR mutation detection (MALDI-TOF MS). It can be provided as a supplementary data.

Response: We appreciate the reviewer’s recommendations. We’ve enhanced the description of EGFR mutation detection method in page 4, line 144-155.

Reviewer 2 Report

The manuscript by Pei-Ya Liao et al. analyzed outcomes and disease progression of patients harboring EGFR-mutant lung adenocarcinoma in relation to radiological severity scores of LM, CSF tumor cell count, and EGFR mutation detection rate in CSF. The important conclusions of the manuscript are that i) the early onset of LM prior to first-line EGFR-tyrosine kinase inhibitor treatment was associated with an independent worse outcome, and ii) the higher radiological severity score of LM predicted higher tumor cell counts in CSF and higher EGFR mutation detection rate. Overall, the manuscript is well constructed; and the data are well presented. Unfortunately, the number of patients is limited, which may impact the conclusion. The manuscript has potential to advance understanding of the prognostic factors of lung adenocarcinoma and the advantages and disadvantages of EGFR-TKI treatment. There are a couple of suggestions regarding the manuscript data and conclusion:

1. The overall survivals of patients with late-onset and early-onset LM were reported as 29.9 months and 12.3 months, respectively (Log-rank P=0.005). Line 221.

This is a striking difference. Do you have data supporting or rejecting the hypothesis that the LM appearance was prognostic for patient survival in both groups? What is the period of the patient survival starting from the first LM detection in both groups (please include this data if it’s available)? 

2. Did you see alterations in the radiological severity score of LM in response to EGFR-TKI treatment in the group with early-onset LM and the group with late LM onset treated with EGFR-TKI?

3. Line 66. Change diver to driver 

4. A list of abbreviations should be included in the manuscript.

I recommend the manuscript for publication in Cancers with minor revision. 

Author Response

  1. Unfortunately, the number of patients is limited, which may impact the conclusion. The manuscript has potential to advance understanding of the prognostic factors of lung adenocarcinoma and the advantages and disadvantages of EGFR-TKI treatment. There are a couple of suggestions regarding the manuscript data and conclusion:

    Response: We appreciate the reviewer’s encouragement and comments. We’ve mentioned. that the major limitation of current study is its retrospective nature and limited patient numbers, and associated descriptions are made in page 10, line 330-336.

  1. The overall survivals of patients with late-onset and early-onset LM were reported as 29.9 months and 12.3 months, respectively (Log-rank P=0.005). Line 221. This is a striking difference. Do you have data supporting or rejecting the hypothesis that the LM appearance was prognostic for patient survival in both groups? What is the period of the patient survival starting from the first LM detection in both groups (please include this data if it’s available)?

Response: We appreciate the reviewer’s comments. As we mentioned in the section of “Method – statistical methods” (page 4, line 161-163), the overall survival we reported was determined as the time from initiation of first-line EGFR-TKI treatment to death of any cause. All patients in our population suffered from leptomeningeal metastasis, and we did not include patients without leptomeningeal metastasis as the controlled group; hence, we cannot make the conclusion that leptomeningeal metastasis was prognostic for NSCLC patients. However, leptomeningeal metastasis is generally associated with a devastating outcome according to the prior studies and we’ve mentioned this point in the section of “Introduction” (page 3, line 81-83) and “Discussion” (page 9, line 263-264) and cited the corresponding references. Herein, we focused on the influence of the timing of leptomeningeal occurrence on patients’ outcome. When we determined the survival time from the first LM detection, the survivals of patients with early onset and late onset LM were 12.5 months (95% CI 4.6-20.4) and 8.5 months (95% CI 4.3-12.6), respectively. Although they are not statistically significant (P = 0.489), patients with late onset LM experienced a numerically shorter survival from LM occurrence to death, which implies the limited treatment options for patients suffering LM after initial EGFR-TKI treatment. We’ve added these data and associated discussion in page 9, line 290-295.

  1. Did you see alterations in the radiological severity score of LM in response to EGFR-TKI treatment in the group with early-onset LM and the group with late LM onset treated with EGFR-TKI?

Response: We appreciate the reviewer’s comments. It is an important and interesting issue to evaluate the prognostic role of dynamic change of biomarkers. However, in the present study, we did not evaluate the alterations in the radiological severity score because the treatments for leptomeningeal metastasis are usually very heterogenous in clinical practice, particularly among those with late onset leptomeningeal metastasis. They had received EGFR-TKI initially; therefore, not all patients underwent EGFR-TKI re-treatment after leptomeningeal metastasis occurrence. Additionally, some of them underwent brain radiotherapy and some did not. In the case of brain MRI, whenever patients had response to treatment, the “leptomeningeal metastasis” lesions may become less obvious but not always disappear, which also confound the calculation of radiological severity score. Prospective studies with large cohort and homogeneous treatment algorithm are required to evaluate the prognostic role of dynamic changes in radiological severity score.  

  1. Line 66. Change diver to driver.

Response: We appreciate the reviewer’s comments. We have revised this typo.

  1. A list of abbreviations should be included in the manuscript.

Response: We appreciate the reviewer’s comments. We’ve provided the list of abbreviations in page 2.

Round 2

Reviewer 1 Report

the authors have improved the manuscript significantly.

Author Response

  1. The authors have improved the manuscript significantly.

Response: We appreciate the reviewer’s encouragement.
